

# Genome annotation across species using deep convolutional neural networks

Ghazaleh Khodabandelou[1,2], Etienne Routhier[1] and Julien Mozziconacci[1,3,4]

[1] Laboratoire de Physique Théorique de la Matière Condensée (LPTMC), Sorbonne Université, Paris, France
[2] Laboratoire Images, Signaux et Systèmes Intelligents (LISSI), Université Val-de-Marne (Paris XII), Paris, France
[3] CNRS UMR 7196 / INSERM U1154 - Sorbonne Université, Museum national d'Histoire naturelle (MNHN), Paris, France
[4] Institut Universitaire de France, Paris, France

## ABSTRACT

Application of deep neural network is a rapidly expanding field now reaching many disciplines including genomics. In particular, convolutional neural networks have been exploited for identifying the functional role of short genomic sequences. These approaches rely on gathering large sets of sequences with known functional role, extracting those sequences from whole-genome-annotations. These sets are then split into learning, test and validation sets in order to train the networks. While the obtained networks perform well on validation sets, they often perform poorly when applied on whole genomes in which the ratio of positive over negative examples can be very different than in the training set. We here address this issue by assessing the genome-wide performance of networks trained with sets exhibiting different ratios of positive to negative examples. As a case study, we use sequences encompassing gene starts from the RefGene database as positive examples and random genomic sequences as negative examples. We then demonstrate that models trained using data from one organism can be used to predict gene-start sites in a related species, when using training sets providing good genome-wide performance. This cross-species application of convolutional neural networks provides a new way to annotate any genome from existing high-quality annotations in a related reference species. It also provides a way to determine whether the sequence motifs recognised by chromatin-associated proteins in different species are conserved or not.

## INTRODUCTION

The improvement of DNA sequencing techniques lead to an explosion in the number and completeness of fully sequenced genomes. One of the major goals in the field is to annotate these DNA sequences, which is to associate a biological function with sequence motifs located at different positions along the genome (*Stein, 2001*). In the human genome for instance, while some DNA sequences encode proteins, most sequences do not code for any protein. Many of these non-coding sequences are nevertheless conserved in related species and are necessary for the correct regulation of gene expression. Deciphering the function of these non-coding sequences has been increasingly achieved through improvements in

Corresponding authors
Ghazaleh Khodabandelou,
ghazaleh.khodabandelou@u-pec.fr,
ghazaleh.khodabandeh@gmail.com
Julien Mozziconacci,
julien.mozziconacci@mnhn.fr

the throughput of next generation sequencing (*Rivera & Ren, 2013*). The 3.2 Billion base pair (bp) long human genome is now annotated with many functional and bio-chemical cues (*Kundaje et al., 2015*; *ENCODE Project Consortium et al., 2012*), among which are the initiation sites of gene transcription (*Carninci et al., 2006*; *Georgakilas, Perdikopanis & Hatzigeorgiou, 2020*). While these annotations are becoming more numerous and precise, they cannot be determined experimentally for every organism and every cell type, as the experiments needed to produce these annotations are often costly and difficult to carry out. Computational methods are therefore widely used to extract sequence information from known annotations and extrapolate the results to different genomes or conditions, e.g., *Kundaje et al. (2015)* and *Durham et al. (2018)*.

An related question is to understand the link between these annotations and the underlying DNA sequence. To this end, supervised machine learning algorithms (*Goodfellow, Bengio & Courville, 2016*) have been particularly successful (*Zou et al., 2019*; *Angermueller et al., 2016*). Among those, deep Convolution Neural Networks (CNNs) are very efficient at detecting sequence features since they rely on the optimisation of convolution filters that can be directly matched to DNA motifs (*Ching et al., 2018*). Stacking several of these convolution layers together can lead to the detection of nested motifs at larger scales. Pioneering studies illustrated this ability of CNNs to reliably grasp complex combinations of DNA motifs and their relationship with functional regions of the genome (*Min et al., 2016*; *Umarov & Solovyev, 2017*; *Alipanahi et al., 2015*; *Zhou & Troyanskaya, 2015*; *Kelley, Snoek & Rinn, 2016*; *Pachganov et al., 2019*).

*Min et al. (2016)* used a CNN to predict enhancers, which are specific sequences that regulate gene expression at a distance. This method performed very well and ranked above state-of-the-art support vector machine based methods. Similar tools were used in different contexts, aiming at identifying promoters (*Umarov & Solovyev, 2017*; *Pachganov et al., 2019*) or detecting splice sites (*Leung et al., 2014*; *Jaganathan et al., 2019*). In these approaches, a sample set is first created by taking all positive class sequences (e.g., enhancers) and adding the same amount of randomly picked negative class examples (e.g., non-enhancers). This sample set is then divided into training, validation and test sets. Balancing the data ensures that the model will be trained on the same number of positive and negative examples, thus giving the same importance to both classes. While these approaches are very successful when assessed on test sets derived from the sample set, we show here that they tend to perform poorly when applied on entire chromosome sequences as required for the task of complete genome annotation. This is due to the fact that the networks are optimised on a similar number of positive and negative examples during training, but that they will usually face very different ratios of negative over positive classes when used on a full chromosome sequence (*He & Garcia, 2009*).

Alternative approaches (*Alipanahi et al., 2015*; *Kelley, Snoek & Rinn, 2016*) used unbalanced datasets for training (i.e., with more negative than positive examples) to predict DNA-binding sites for proteins and genome accessibility. In these two studies, however, the prediction performance of the model is also assessed on test sets derived from training sets, not on full genomic sequences. The task of genome-wide prediction has been assessed in a more recent study aiming at identifying cell type specific regulatory

elements (*Kelley et al., 2018*). In order to infer long range relationships between these elements, Kelley et al. used very long (131 kb) non-overlapping windows covering the whole genome. This approach has proven efficient but requires a lot of computational memory.

As our goal is to provide genome-wide predictions, the methodology we used is inspired from this last study. Since we do not aim here at predicting cell type specific features, we could use shorter sequences as input and a simpler network architecture. We also present two novelties for the development and for performance assessment of genome-wide predictions. Firstly, we do not use as a quality measure the classical prediction scores computed on test sets obtained by dividing the sample data into training, validation and test sets as commonly done in machine learning. Rather, we compute prediction scores that assess the ability of our model to annotate a full chromosome sequence by designing a specific metric (described in Material and Methods). Secondly, we change the ratio between positive and negative examples in order to obtain the highest prediction scores and show that this tuning is has an important effect on the outcome. As a proof of principle, we use in this work gene start sites (GSS) as features. DNA motifs around GSS are recognised by the transcription machinery and indicate the location of the initiation of transcription (*Kugel & Goodrich, 2017*). The DNA sequence surrounding GSS therefore contains the information that could in principle be used by an algorithm to identify in silico the GSS locations. These DNA sequence motifs are different for different classes to genes. For instance, protein coding genes can have either CG di-nucleotide (CpG) rich or poor sequences upstream their GSS (*Deaton & Bird, 2011*). We show that using training sets with a higher ratio of negative over positive examples, we can faithfully retrieve GSS positions, with performances varying for different classes of genes such as coding or non coding genes.

We then propose a new application of CNNs in genomics that leverages the fact that similar organisms tend to have similar regulatory mechanisms, i.e., rely on homologous molecular machinery and on homologous DNA regulatory motifs. Exploiting these homologies, we first train a model on a dataset corresponding to a given organism and use it to predict the annotation on the genome of a related organism, opening new opportunities for the task of *de-novo* genome annotation. We show that a CNN trained on GSS containing regions in human is able to recover regions containing GSS in the mouse genome and vice versa. We also assess the generalisation of the approach to more distant species, taking as examples *Gallus gallus* and *Danio rerio*.

## METHODS

### Input generation

Genomic sequences were downloaded for the reference genomes for Human (hg38), Mouse (mm10), Chicken (gg4) and Zebrafish (dr10) via the URLs shown in Table 1. Similarly, GSS positions for each genome were extracted from their respective NCBI RefSeq Reference Gene annotations (RefGene).

As a positive input class, we use regions of 299 bp flanking GSS (i.e., ±149 bp around the GSS) which are supposed to contain multiple sequence signals indicating the presence

**Table 1  URL of the data used in the present work.**

**Genomes**

| | |
|---|---|
| human | https://hgdownload.soe.ucsc.edu/goldenPath/hg38/bigZips/hg38.fa.gz |
| mouse | https://hgdownload.soe.ucsc.edu/goldenPath/mm10/bigZips/chromFa.tar.gz |
| chicken | https://hgdownload.soe.ucsc.edu/goldenPath/galGal4/bigZips/galGal4.fa.gz |
| zebrafish | https://egg.wustl.edu/d/danRer10/refGene.gz |

**Reference Gene**

| | |
|---|---|
| human | https://egg.wustl.edu/d/hg38/refGene.gz |
| mouse | https://egg.wustl.edu/d/mm10/refGene.gz |
| chicken | https://egg.wustl.edu/d/galGal4/refGene.gz |
| zebrafish | https://egg.wustl.edu/d/danRer10/refGene.gz |

of a GSS to the transcription machinery of the cell. For instance in the human genome, 31,037 GSS positions are extracted on both DNA strands (15,798 for the positive strand and 15,239 for the negative strand). In order to generate the negative class, we select $31,037 \times Q$ sequences of 299 bp at random positions on a random strand, rejecting regions that do contain a GSS. The odds of getting at random a genomic region containing a GSS are close to 0.28%. For $Q = 1$, there is an equal number of negative and positive class examples. Unbalanced datasets are produced using different values of $Q$ ranging from 1 to 100. For $Q = 100$, the negative class encompasses $100 \times 299 \times 31k \approx 1Gb$, which represents one third of the human genome. For the other genomes a similar procedure was implemented. The total number of GSS used was 25,698 for the mouse, 6876 for the chicken and 14805 for the zebrafish.

## Convolution Neural Network (CNN)

A CNN (see Fig. 1) is trained in order to predict the presence of a GSS in a DNA sequence of size 299 bp. The shape of the input layer is $c \times b$ in which $c = 4$ is the number of different nucleotides and $b = 299$ is the length of the input sequence. The nucleotide sequences are one hot encoded so that A=(1,0,0,0), T=(0,1,0,0), C=(0,0,1,0), and G=(0,0,0,1). The training set contains $N$ samples of labelled pairs $(X^{(n)}, y^{(n)})$, for $n \in \{1,\ldots,N\}$, where $X^{(n)}$ are matrices of size $c \times b$ and $y^{(n)} \in \{0,1\}$. Each $X^{(n)}$ is associated with $y^{(n)} = 1$ when it corresponds to a region containing a GSS and $y^{(n)} = 0$ otherwise. The first convolution layer consists of $k$ kernels of length $s$ which are applied on $b - s + 1$ successive sequences at positions $p \in \{1,\ldots,(b-s+1)\}$ to recognise relevant DNA motifs of size $s$. This operation generates an output feature map of size $k \times (b-s+1)$ for an input $X^{(n)}$ of size $c \times b$. The feature map $\mathcal{M}$ resulting from the convolution operation is computed as follows:

$$\mathcal{M}_{p,i} = \sum_{j=1}^{c} \sum_{r=1}^{s} \mathcal{W}_{i,j,r} X_{p+r-1,j} + \mathcal{B}_i, \quad i \in \{1,\ldots,k\} \tag{1}$$

where $\mathcal{W}$ denotes the network weights with size $(k \times c \times s)$ and $\mathcal{B}$ denotes the biases with size $(k \times 1)$ (see e.g., *Goodfellow, Bengio & Courville, 2016*). After the convolution layer a non-linear function is applied to the output, here a Rectified Linear Unit (ReLU). This

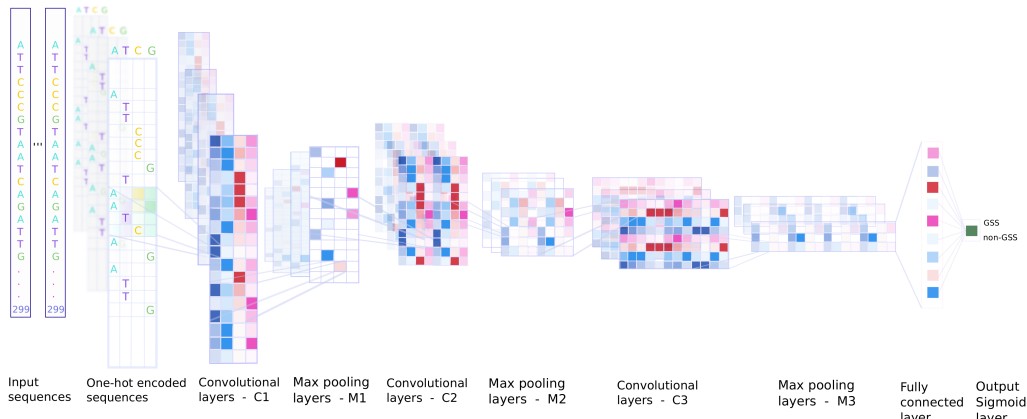

**Figure 1 Overview of the CNN model.** A total of 299 bp-long sequences were one hot encoded into a $4 \times 299$ input matrix. The first CNN layer performs a convolution on each input matrix to recognise relevant motifs. The next convolutional layers models the interplay among these motifs to grasp higher-level features. Max-pooling layers reduce the dimensions of the layers. The model is trained to correctly label input sequences as GSS or non-GSS. The output layer of the trained network then gives a probability for any 299 bp region to contain a GSS. It can be applied along a full chromosome, i.e., on all 299 bp-long sequences with a 1 bp shift.

activation function computes $f_{ReLU}(\mathcal{M}) = max(0, \mathcal{M})$ to incorporate non-linearity by transforming all negative values to zero. In order to reduce the input dimension we apply a max-pooling process with a pool size $m$ over the output of $f_{ReLU}(\mathcal{M})$. Similar convolution layers followed by ReLu and max-pooling are added sequentially on the input of the first layer to grasp higher order motifs. The output of the last max-pooling layer is then fed into a fully connected layer which output $x$ is transformed by a softmax layer, i.e., a sigmoid function ($\phi = \frac{1}{1+e^{-x}}$), in order to give the final output of the CNN. This final score of the input sequence is ideally 0 for non-GSS and 1 for GSS containing sequences. When we need to perform a classification we use a threshold of 0.5 to discriminate between the two classes.

In the training phase, the weights and biases of the convolution layers and the fully connected layer are updated via back-propagation (*Rumelhart, Hinton & Williams, 1986*) in a way which decreases the loss, which measures the discrepancy between the network predictions and the reality averaged over individual examples. We use here the binary cross-entropy computed as:

$$\mathcal{L} = -1/N \sum_{i=1}^{N} [y^{(n)} log(\hat{y}^{(n)}) + (1 - y^{(n)}) \times log(1 - \hat{y}^{(n)})] \tag{2}$$

where $\hat{y}^{(n)}$ is the estimated score for the input sample $X^{(n)}$.

As data are imbalanced for $Q > 1$, the model may reach a local optimum when predicting the non-GSS class for all input sequences. In order to deal with this issue, we attribute different weights to the positive and negative classes. We assign a greater importance to the less represented GSS class by multiplying the associated term in the loss by a weight $CW = \frac{\text{number of non-GSS}}{\text{number of GSS}} = Q$.

One of the important issues of any learning algorithm is overfitting. Overfitting occurs when one achieves a good fit of the model on the training and validation data, while it does not generalise well on new, unseen data. To deal with this issue, a regularisation procedure called dropout is usually used (*Srivastava et al., 2014*). In the training step, some outputs of the pooling layers are randomly masked while the remaining information is fed as inputs for the next layer.

## Implementation

We implement CNN using Keras library (*Chollet et al., 2015*) and Tensorflow (*Abadi, Agarwal & Barham, 2015*) as back-end. Training on a GPU is typically faster than on a CPU. We use here a GTX 1070 Ti GPU. We use Adaptive Moment Estimation (Adam) to compute adaptive learning rates for each parameter (*Kingma & Ba, 2014*). Adam optimiser is an algorithm for first-order stochastic gradient-based optimisation of functions, based on adaptive estimates of lower-order moments. The network architecture (see Fig. 1) is detailed in Table 2. The models are trained for 150 epochs and they mostly converge rapidly (around 30–35 epochs, we use early stopping to prevent overfitting). Hyper-parameters tuning is detailed in the supplementary materials.

Source codes are available at https://github.com/StudyTSS/DeepTSS/.

## Genome wide performance measure

Different measures have been developed in order to assess the performance of different models on conventional test sets, i.e., test sets derived from a subset of the initial data. Such measures are described in details in the corresponding supplementary materials section. In our case, we want to apply our model on all the 299 bp windows spanning a full chromosome and eventually chromosomes from other species. Specifically, the model was tested on chromosome 21 which was withdrawn from the training set. We therefore developed a measure to evaluate the performance of the trained models in this case. This metric, called $\lambda$, measures the enhancement of the predicted signal specifically in the regions surrounding the known GSS. We use in the present papers regions of length $r = 400$ bp. To compute $\lambda$, we first compute the genome-wide Z-score (*Kreyszig, 2009*) $Z_g = \frac{y_g - \bar{\mu}}{\sigma}$ from the predictions $y_g$ where $g$ denotes positions on the genome, and $\bar{\mu}$ and $\sigma$ stand for the prediction mean and standard deviation, respectively. We extract $\mathcal{Z}_{GSS}$, the $Z_g$ signal over 10 kb windows centred on each GSS of the test region, e.g., a full chromosome. $Z_g$ is a 2D-array whose rows correspond to different genes and columns to different distances to the GSS. We then average element-wise $\mathcal{Z}_{GSS}$ over all GSS, i.e., along all rows. This gives us $S$, the average of the Z-transformed prediction score in a 10 kb window around all GSS. In order to measure the signal increase close to the GSS, that we call $\lambda$, we compute the average of the curve $S$ on a region of $r$ bp centred on the GSS. A higher value of $\lambda$ corresponds to a higher signal-to-noise ratio around the GSS.

**Table 2 Network architecture of the CNN model.** The first column depicts the different layers used consecutively in the network. The "layer shape" column reports the shape of the convolutional kernels, the max-pooling windows and the fully connected layers. The "output shape" column reports the variation of layer shapes at each step.

| Layer name | Layer shape | Output shape |
| --- | --- | --- |
| Input | – | $4 \times 299 \times 1$ |
| Conv2D | $32 \times 4 \times (4 \times 1)$ | $32 \times 296 \times 1$ |
| Max-pooling | $2 \times 1$ | $32 \times 148 \times 1$ |
| Dropout | – | $32 \times 148 \times 1$ |
| Conv2D | $64 \times 32 \times (4 \times 1)$ | $64 \times 145 \times 1$ |
| Max-pooling | $2 \times 1$ | $64 \times 72 \times 1$ |
| Dropout | – | $64 \times 72 \times 1$ |
| Conv2D | $128 \times 64 \times (4 \times 1)$ | $128 \times 69 \times 1$ |
| Max-pooling | $2 \times 1$ | $128 \times 34 \times 1$ |
| Dropout | – | $128 \times 34 \times 1$ |
| Dense | 128 | 128 |
| Dropout | – | 128 |
| Dense (sigmoid) | 1 | 1 |

## RESULTS

### Training models for genome annotation of GSS

The problem of detecting human GSS using deep neural networks has been tackled in (*Umarov & Solovyev, 2017*). We first follow a similar approach and use a balanced dataset (see Methods for details). The model is trained/validated on an equal number of 299 bp long positive and negative examples and is evaluated on a test set composed of 15% of the original data that was left aside prior to training. The specificity (Sp), the sensitivity (Sn) and the Matthews Correlation Coefficient (MCC, *Chicco & Jurman, 2020*) (see Supplemental Information 1 for definition) were found to be similar to the ones found in (*Umarov & Solovyev, 2017*) which used a similar approach albeit separating the sample data into TATA-containing GSS and non-TATA GSS (Sp = 0.94, Sn = 0.92 and MCC = 0.86).

In order to assess how this model would perform as a practical tool for detecting GSS on a genome-wide scale, we apply it on all the sequences along chromosome 21 (which has been withdrawn from the training phase, i.e., from the training and validation sets) obtained using a 299 bp long window sliding with an offset of 1 bp. Figure 2A illustrates the predictions of the CNN model over a typical region of 300 kbp containing 7 out of the 480 GSS of chromosome 21. Although the predictions yield higher scores over GSS positions, they also yield high scores over many non-GSS positions reflecting a low signal-to-noise ratio. This is due to the fact that the reality is biased in the training phase during which the CNN model learns an equal number of examples from the positive and the negative classes (*He & Garcia, 2009*). Applied over all the 299-bp sequences of chromosome 21, the model encounters many more examples of the negative class and fails to generalise to the new examples.

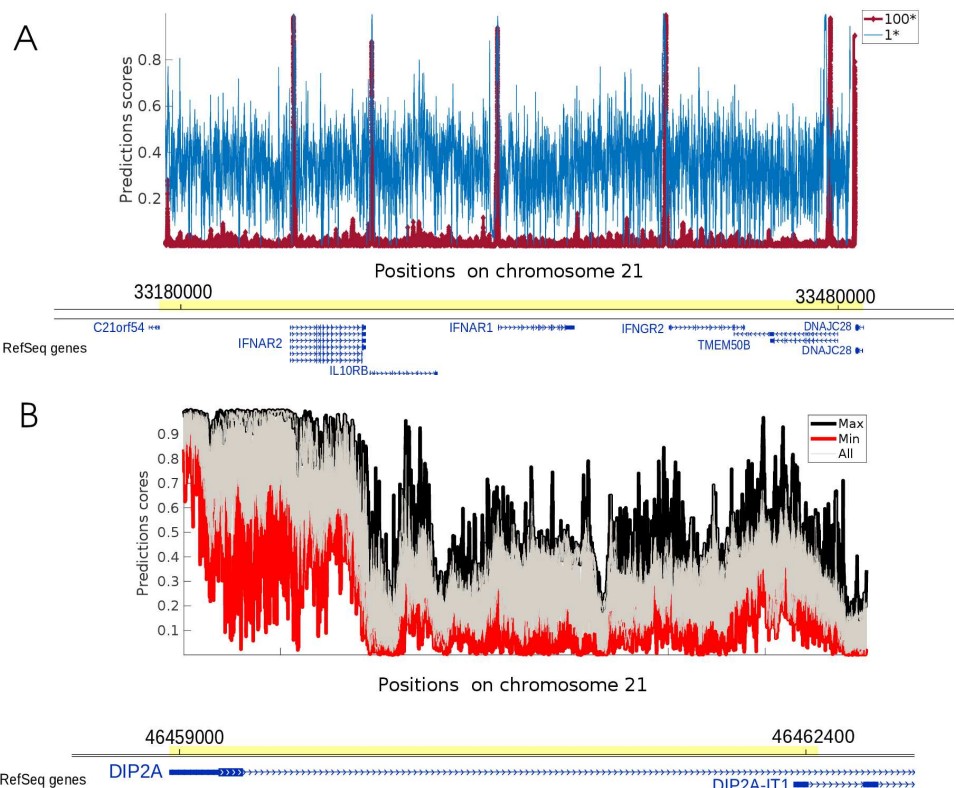

**Figure 2** **CNN predictions for two regions of chromosome 21.** (A) Prediction scores for balanced 1⋆ model ($Q = 1$) and unbalanced 100⋆ model ($Q = 100$), respectively in blue and red on a 300 kb region. The position of genes is indicated below. The annotation track was done using the UWash Epigenome browser (https://epigenomegateway.wustl.edu/). Both models detect 7 GSS positions, but the 1⋆ model returns a higher background signal at non GSS positions. Adding negative examples using the 100⋆ model mitigates the noise while preserving the high scores over GSS. (B) Application of 30 1⋆ models, trained on different datasets, over a 3.2 kb region of chromosome 21. At each site, the maximum and minimum prediction scores are respectively displayed in black and red. Other prediction scores are plotted in grey.

To address this issue and train a network for genome annotation, we propose a heuristic where more negative examples are added into the balanced dataset to reduce the importance of the positive class during training and to allocate more weight to the negative class. We call these augmented datasets *limited unbalanced* datasets. The parameter $Q$ is the ratio between negative and positive training examples and denote as $Q^*$ models trained with the corresponding ratio. For instance, on Fig. 2A the model trained on the balanced data yielding to blue signal predictions is denoted as 1∗. We train our CNN model on a 100⋆ dataset ($Q = 100$) and assess the efficiency of the trained model. As depicted on Fig. 2A by a red signal, the predictions for this model display a much higher signal-to-noise ratio, with significant peaks over each of the 7 GSS (C21orf54, IFNAR2, IL10RB, IFNAR1, IFNGR2, TMEM50B, DNAJC28) and a much weaker signal between these sites. Predicting GSS using the 100⋆ model is thus expected to generate fewer false positives than the 1⋆ model, regardless the value of the threshold used to identify GSS-containing regions. In order to

assess how changing the value of $Q$ affects GSS classification, we apply a threshold on the prediction and compute the precision and the recall obtained for both models (i.e., 1* and 100*) at 600 bp resolution on a full chromosome. The precision recall curves confirmed the compromising effect of a lower signal-to-noise ratio on the accuracy of the classification (Fig. S1). For the sake of completeness, the performance of more models (1*, 10*, 20*, 30*, 50*, 100*) evaluated using conventional metrics on test sets derived from the initial sample sets can be found in Supplemental Information 1.

## Investigating the effect of random selection of the negative examples on predictions

While positive examples are always the same in different sample sets, the negative examples are randomly picked out of the genome. The performance of the model in different regions of chromosome 21 can thus vary for different training sets (*Wesolowska-Andersen et al., 2020*). To investigate this variation, we set up 30 balanced 1* datasets and train 30 CNNs separately. The 30 models are then applied over human chromosome 21 to study the fluctuations of the predictions. The variation of 30 predictions is depicted in Fig. 2B. The first observation is that almost all predictions present a peak over the DIP2A GSS. However, the large gap between the minimum and maximum predictions underlines the variability of predictions obtained with different training datasets. This variability illustrates the uncertainty of the predictions obtained from a single CNN trained on a balanced dataset and highlights the need to use limited unbalanced datasets for the task of genome annotation.

## Comparing 1* and 100* models over a full chromosome

Models trained on 1* and 100* sets are applied to the full chromosome 21 and the Z-normalized prediction scores around GSS are presented as heat-maps. While the 1* model (Fig. 3A) presents a noisy signal around GSS positions, the 100* model (Fig. 3B) presents a higher signal-to-noise ratio. To investigate the performance of different models on a genome-wide scale we devised a custom metric $\lambda$ which measures the average signal-to-noise ratio around GSS (see Methods for the definition of $\lambda$).

Figures 3C, 3D illustrate the average of the Z-score over all the GSS of chromosome 21 for the models 1* and 100*, respectively, and $\lambda$ denotes the average of this average over a $r = 400$ bp region centred on the GSS. A larger $\lambda$ score corresponds to a higher signal-to-noise ratio. In this particular case, we find a $\lambda$ score of 2.21 and 5.81 for the 1* and 100* model, respectively.

To illustrate the variability of prediction scores achieved around different GSS, we randomly selected four GSS within the chromosome. The first GSS corresponds to the gene CXADR, shown in Fig. 3E. While the prediction of 1* model results in a low averaged Z-scores over all positions, the averaged Z-score of 100* model strongly peaks around the GSS position and shows low variations over non-GSS positions. Figure 3F depicts the second selected GSS corresponding to the KRTAP19-2 gene. This gene is part of a cluster of similar genes belonging to the family of Keratin Associated Proteins (highlighted by a yellow rectangle on Figs. 3A, 3B). For this particular cluster, the predictions are poor for

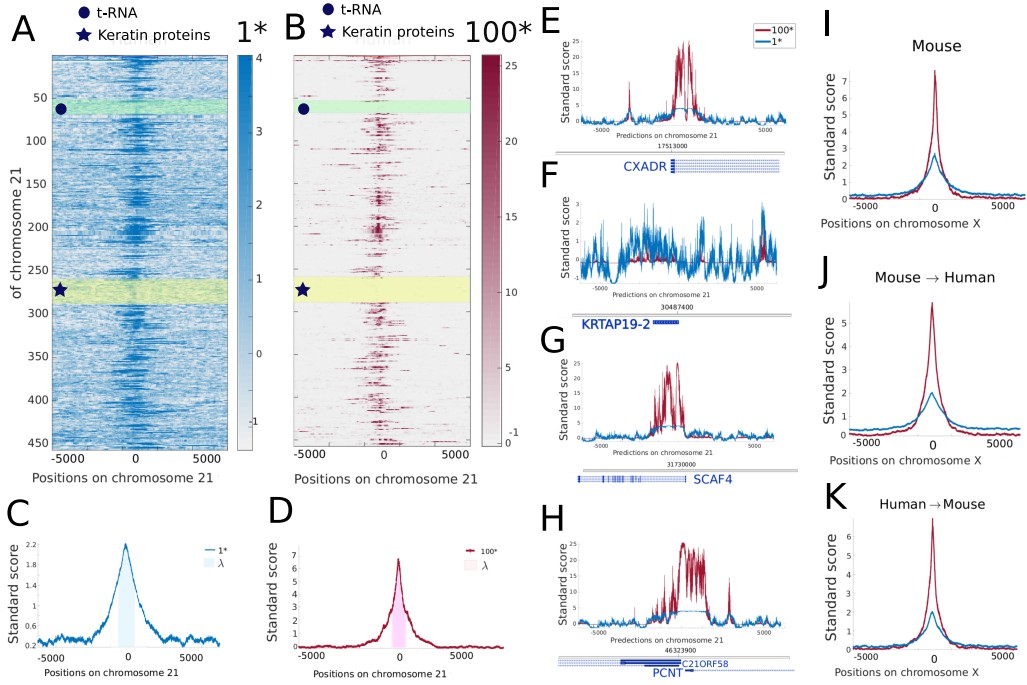

**Figure 3  Comparison of the 1* and 100* models predictions over chromosome 21.** (A) and (B) Heat maps depict the Z-score of the prediction for the 1* and 100* models respectively on 5,000 bp flanking each GSS of chromosome 21. (C) and (D) Averaged Z-score of the predictions over each GSS of chromosome 21. (E–H) Zoom on regions around randomly selected GSS. Genes are indicated at the bottom of each plot. (I–K) Averaged Z-score of the predictions over each GSS of mouse chromosome X (I) and for networks trained on mouse/human chromosomes (except X) and applied on human/mouse chromosome X (J,K).

both 1* and 100*, probably reflecting a specific GSS signature that has not been grasped by the model. Another example of gene cluster with a poor prediction score for GSS is the t-RNA cluster, highlighted in green in Figs. 3A, 3B. Figures 3G, 3H displays the predictions around the GSS of the SCAF4 and, PCNT and C21ORF58 genes, respectively. On these more typical GSS the 100* model shows a higher signal-to-noise ratio than the 1* and regions containing GSS are detected. These regions often stretch over 1 kb while our training sequence centred on each GSS is only 299 bp long. This could indicate the presence either of alternative GSS close to the annotated GSS or of similar sequence patterns in broader regions surrounding the GSS (*Carninci et al., 2006*; *Sandelin et al., 2007*).

## Learning and predicting in human and mouse

To show the potential of our annotation method in a different context, we replicate a similar GSS analysis in mouse. Models with values of $Q$ ranging from 1 to 100 trained on mouse chromosomes (except X) are applied over the mouse chromosome X to assess the model performance (see Fig. 3I, and Figs. S2A, S2D, S2G). The averaged Z-score of $\lambda$ reaches values of 2.24 and 4.90 respectively for the 1* and 100* models in quantitative agreement with the model performance in human.

Mammals show a substantial degree of homology in the DNA sequence found at GSS (*Waterston et al., 2002*), and earlier computational models were trained to recognise transcription start site in any mammalian species (*Down & Hubbard, 2002*). This study focused on 313 sequences, of which 50 were kept aside for test purposes and we want here to extend the validity of this initial study at the genome wide level. Following this line, we determine the possibility of predicting GSS in one organism with a network trained on a related organisms. This possibility has previously been shown to be effective for sequence variants calling (*Poplin et al., 2018*) To this end, the mouse trained model is applied on human chromosome X and the human trained model is applied on mouse chromosome X. The two chromosomes carry homologous genes (*Waterston et al., 2002*), the number of annotated GSS varies with a total of 4,968 GSS in human and 2,005 GSS in mouse. While the model trained and applied on mouse shows a better signal-to-noise ratio, the same model applied to human chromosome X still captures most of the GSS and gives a $\lambda$ score of 5.18 for the 100* model (see Fig. 3J and Figs. S2B, S2E, S2H). Similarly, the models trained on human capture most of GSS on the mouse X chromosome as shown in Fig. 3K and Figs. S2C, S2F, S2I and reaches a $\lambda$ score of 4.32 for the 100* model. In all cases, the signal-to-noise ratio is improved in the 100* models with respect to the 1* models. The human model applied on human provides the highest scores for both 1* and 100* models probably a signature of an overall better GSS annotation.

## Evaluation of the prediction for different GSS classes

The potential of our trained networks to recover GSS containing regions along the human and mouse genomes is assessed in the previous parts without any distinction between different GSS classes. Since we find that some GSS are better predicted than others (Fig. 3), we compute the $\lambda$ score independently for the two main classes of GSS: mRNA-GSS and ncRNA-GSS. While $\lambda$ is higher for the mRNA-GSS class, the model is versatile and is also able to predict the ncRNA-GSS (Fig. 4B). In human and mouse, mRNA-GSS are found in different classes, that can be derived from the CpG content of the region flanking the GSS. High CpG regions, also called "CpG island" can be methylated and play an important role in gene regulation (*Deaton & Bird, 2011*). Figure 4A displays the distribution of the CpG number in 299 bp regions surrounding the all mRNA-GSS for the mouse and human X chromosome. From this distribution, we identify three classes of mRNA-GSS with respectively a high, medium and low CpG content. High CpG GSS correspond to genes regulated by DNA methylation and have been shown to exhibit a different pattern of chromatin modifications (*Vavouri & Lehner, 2012*). Assessing the performance of the model for the three different classes, we find that better scores are obtained for CpG richer GSS (Fig. 4B). The worst performing GSS are low CpG content GSS which are hardly recovered by our model. In order to test whether CpG content alone could be used to predict GSS we computed the $\lambda$ score over all GSS using the Z-normalized CpG content as predictor. We get values of 1.30 and 0.92 respectively for the human and mouse GSS indicating that the CpG content is a strong indicator of the present of GSS but that our models use as well other features which allow them to reach much higher scores.

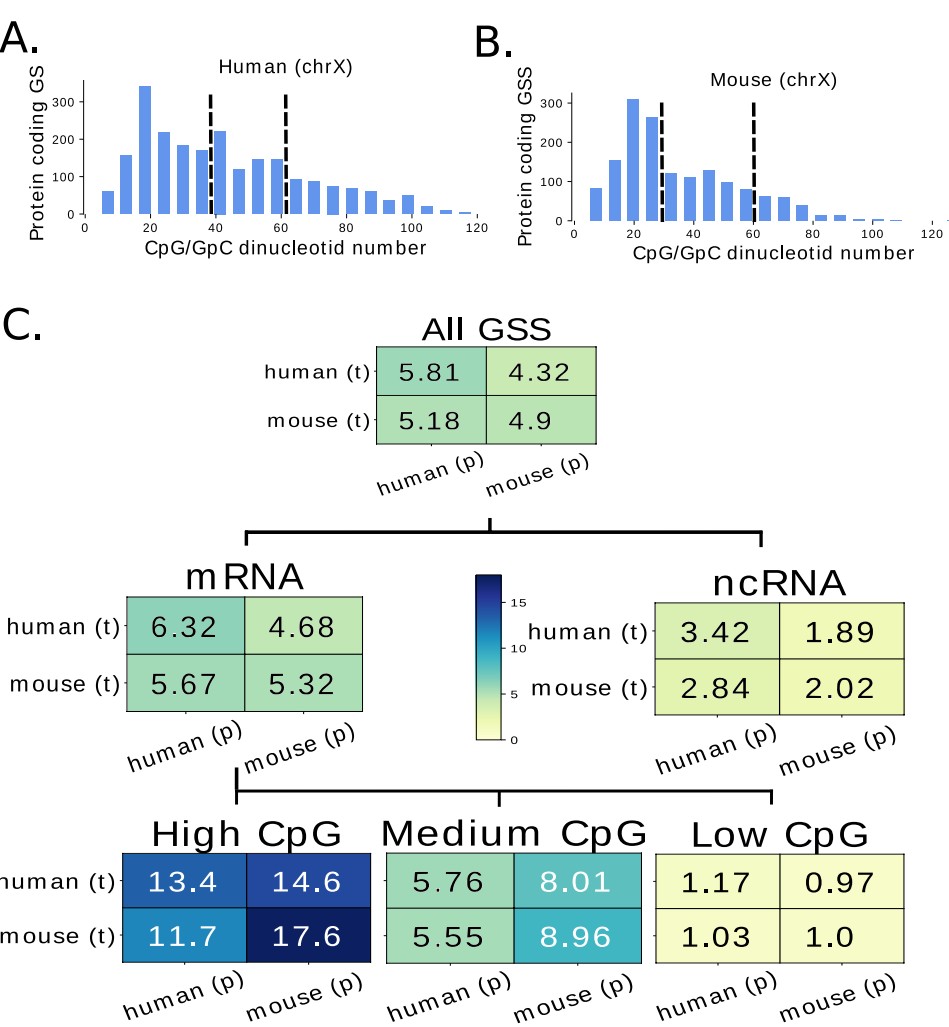

**Figure 4 Evaluation of the model performance for different classes of genes.** (A) and (B) CpG number in 299 bp regions centred on mRNA-GSS in X chromosomes for human (A) and mouse (B). These regions were divided in three groups of similar size according to their CpG number into low, medium and high groups (the bounds are 35% and 60% for human and 30% and 60% for mouse). The proportion of genes in each class is similar on the X chromosome (test set) than on other chromosomes (training and validation sets). (C) Lambda values computed for networks trained on each species non-X chromosome GSS (t) and predicted on either species' X-chromosome GSS (p). Lambda values for each mRNA-CpG sub-group and ncRNA genes are also shown to highlight different levels of performance.

## Application of the approach to other vertebrates

The performance of a CNN trained on human GSS to recover mouse GSS is not surprising given the similarity between their genomes (*Waterston et al., 2002*). We next set out to apply the same methodology on more diverse species, including chicken and zebrafish (Fig. 5). Four CNNs were trained on all the GSS from the genomes of *Homo Sapiens* (human), *Mouse musculus* (mouse), *Gallus gallus* (chicken) and *Danio rerio* (zebrafish). G.g. and D.r. are model organisms, and together with H.s. and M.m. provide the most comprehensive GSS annotations for mammals, birds and fishes. These four CNNs were

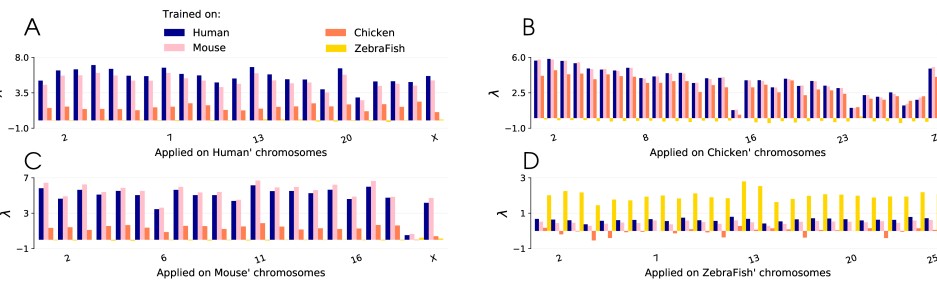

**Figure 5** **Lambda scores obtained with CNN trained on four different species: human, mouse, chicken and zebrafish.** Lambda scores are computed from predictions done on GSS of (A) human, (B) mouse, (C) chicken and (D) zebrafish chromosomes.

then applied genome wide on each of the four species and the λ metric is computed for each chromosome independently, using a *r* value of 400 bp (see Methods).

The results for the human and mouse genomes are very similar, with only a slightly better performance when the model trained on a species is applied on the same species. The model trained on the chicken genome performs less well when applied on the mammalian genomes and the model trained on the zebrafish genome is not able recover the mammalian GSS as shown by a λ value of 0.

When applied on the chicken genome, the mouse and human models surprisingly outperform the chicken model, probably because the GSS annotation is better in the two mammals so that the training phase is more efficient. This result highlights the potential of the method when used across different species when the genome of one species is more precisely annotated.

When applied on the zebrafish genome on the other hand, the human, mouse and chicken models all show poor performances while the zebrafish model performs well. This is in line with the fact that the CpG composition of zebrafish regions around GSS if very different than in chicken and mammals. CpG islands, which are high density CpG regions, are found upstream of many GSS for coding genes in chicken and mammals while they are less abundant in the zebrafish's genome which has a low GC content (*Han et al., 2008*). All together, these results suggest that the molecular machinery that interprets the genome sequence in order to find start sites of genes has a similar specificity in human, mouse and chicken, but a different specificity in zebrafish.

## CONCLUSIONS

With the surge of DNA sequencing technologies, over a million genome datasets are now available and petabases of transcripts are sequenced every year to annotate these datasets with functional marks (*Wainberg et al., 2018*). It has not escaped the notice of many computational biologists that deep neural networks are a key tool to deal with this exponentially increasing amount of data (*Wainberg et al., 2018*). One possible application is to leverage datasets with good annotations in order to train neural networks and to predict annotations on other datasets. One of the practical issues when applying neural networks

on genomic sequences is unbalanced data, a well-known issue in the machine learning literature (*He & Garcia, 2009*; *Chawla, Japkowicz & Kotcz, 2004*; *Batista, Prati & Monard, 2004*). In the present paper, we address this problem using GSS as a case study. Indeed, GSS occupy only a few locations on the genome (31,037 GSS for human) leading to extreme unbalances in datasets (i.e., the ratio of GSS-containing 299 bp windows to non-GSS in the human genome is 1/400). In this case, the lack of examples of the minority class (i.e., true GSS) impacts the learning process as conventional machine learning algorithms usually measure the model performance on the majority class (i.e., non-GSS) leading to biased or inaccurate prediction of the minority class. To deal with this disparity, we adopt a weighting strategy to decrease the importance of the majority class samples (non-GSS) during the learning process thereby improving identification of the rare minority class samples (GSS). Using this approach, which we call "limited unbalanced datasets", we show that learning on imbalanced datasets can be performed effectively, and that for GSS recognition, a ratio of 1 to 100 positive over negative examples is usually sufficient to achieve a good signal to noise ratio in the prediction. This approach can be easily extended to identify other functional regions in any annotated genome.

We also show that our method can be efficiently used across genomes of different species, i.e., training the model on one genome and applying it to another genome. We use the X chromosomes of human and mouse GSS as a case study, and apply models trained on each one's other chromosomes to its own and the other one's X chromosome. While the sequence of this chromosome has evolved differently in both species, many genes are homologous (*Sinha & Meller, 2007*). The fact that we are able to recover GSS in mouse/human with a model trained on the other organism suggests that the machinery capable of recognising GSS in each organism is overall conserved. We also show that this methodology can be applied to more distant species, and use as examples chicken and zebrafish. Our results point toward a higher similarity between mammal and chicken while zebrafish GSS cannot cannot be reliably predicted with models trained on mammal and chicken sequences. While the genome sequence conservation can be computed directly from DNA sequences, further developments of our method may provide a new tool to quantify more complex patterns of similarity between different organism's nuclear machinery that interprets DNA sequences *in vivo*.

## ACKNOWLEDGEMENTS

We would like to thank Léopold Carron for helping us with datasets, Hugues Roest Croeluis for discussions, Michel Quaggetto for technical support and Annick Lesne for comments on the manuscript. We also wish to thank our editor James Procter and the two anonymous referees for their invaluable work.

### Funding
This work was supported by the Agence Nationale pour la Recherche [HiResBac ANR-15-CE11-0023-03]. The funders had no role in study design, data collection and analysis, decision to publish, or preparation of the manuscript.

### Grant Disclosures
The following grant information was disclosed by the authors:
Agence Nationale pour la Recherche [HiResBac ANR-15-CE11-0023-03].

### Competing Interests
The authors declare there are no competing interests.

### Author Contributions
- Ghazaleh Khodabandelou conceived and designed the experiments, performed the experiments, analyzed the data, performed the computation work, prepared figures and/or tables, authored or reviewed drafts of the paper, and approved the final draft.
- Etienne Routhier performed the experiments, performed the computation work, prepared figures and/or tables, and approved the final draft.
- Julien Mozziconacci conceived and designed the experiments, analyzed the data, authored or reviewed drafts of the paper, and approved the final draft.

### Data Availability
Code is available at GitHub: https://github.com/StudyTSS/DeepTSS.

### Supplemental Information
Supplemental information for this article can be found online at http://dx.doi.org/10.7717/peerj-cs.278#supplemental-information.

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
