# Peer review of "Genome annotation across species using deep convolutional neural networks"

_PeerJ Computer Science, doi:10.7717/peerj-cs.278_

## Round 0.1 · original submission · Major Revisions

Dear Professor Mozziconacci,

Thank you for your email regarding my editorial decision concerning your manuscript on genome functional annotation with deep CNN.

I was pleased that you feel able to address the majority of the reviewer's points.

From your email, I now also understand better your intention regarding the application of TSS predictors across different species; indeed, I am sure you are aware of the earlier work done in this field, such as Carninci et al. (https://www.nature.com/articles/ng1789). Whilst there are some conceptual similarities with the canine morphology analogy you described in your email, I think it is important that in your revised submission you more fully explain to the reader why differences in TSS may occur and give at least some account of prior work in this regard - perhaps even going as far to report the performance of existing TSS predictors for this problem, so your method's performance can be evaluated more thoroughly.

As I stated in my original decision, an account of why deep-CNNs may be a more effective tool than existing predictors will also be extremely useful to the computational biology community.

In your email you suggest that the data augmentation approach can be simply omitted. My only concern here was that this aspect of the work seems quite important for the performance of the method, and appeared prominently in the abstract. Simply omitting it entirely (as you intend) seems drastic, since R1's comments (below) suggest that the augmentation approach was simply not correctly implemented, rather than being fatally flawed:
> "The [data augmentation] approach is biased by the fact that similar
> sequences, albeit with different offsets... are found in both training and
> test sets". This makes any evaluation of the model on the validation or
> test set worse than useless as the model can achieve perfect scores by
> overfitting to the training data. Data augmentation should always be
> performed after data has already been split into training, test and
> validation subsets, to prevent contamination. I think that this need to be
> corrected as it is a major experimental flaw.

My reading of this suggests augmentation can be applied *after* training data selection (in the same way as any other data synthesis method), since the leakage referred to is simply because the same TSS locus appears in training and test data. My hope here is that it will be possible for your colleagues to repeat the experiments to evaluate whether the correctly implemented augmentation strategy is effective. Please also note R1 and R2 both comment on the way in which the 'dataset imbalance' problem is introduced and analysed through your computational investigations.

Thanks again for your perserverence regarding my decision. I look forward to receiving your revised manuscript in due course.

Sincerely,
Jim Procter.

· Appeal

Appeal

Dear Dr Procter,

I am Julien Mozziconacci, senior corresponding author for this paper.

I would like to thank you for your time as an editor and for the

insights you brought together with the two referees.

First of all we apologize for the typographical and grammatical errors

that could be present in the manuscript. We will make a huge effort to

find them and we acknowledge the tremendous work of R2 which will help

us a lot.

On top of these considerations on the form of the manuscript, two main

scientific issues were raised by R1.

- The first is about data augmentation. We agree with R1 on this point

and we will thus remove this minor part of the manuscript on data

augmentation (which only concerns one paragraph of supplementary

discussion and one supplementary figure which are unnecessary to

convey our message).

- Secondly, it seems nonsense to R1 to train a network in one specie

and apply it on another one. While I can understand the point he

makes, our results in fact suggest the opposite, raising great

perspectives for the field. In my eyes this is an important result of

the paper. Following the dog metaphor of the referee, we show here

that a convolution neural network trained to recognise dogs using only

pictures of labradors, DOES correctly identify pictures of other dogs,

such as rottweiller or border collie but is not able to correctly

label pictures of poodle. In our case, we show that human TSS are very

alike mouse and chicken TSS but greatly differ from fish TSS.

At this point, we would like to have your opinion. We feel confident

to be able to satisfy all reviewers request excepting the second point

raised by R1. In this case, would you consider a re-submission of our

paper in PeerJ ?

Best wishes and happy new year,

Julien Mozziconacci


· · Academic Editor

Reject

Two reviewers have evaluated your manuscript. They have identified a number of concerns: R1 specifically highlights potentially serious issues concerning the data augmentation strategy, and poses questions regarding the likely success of any TSS predictor trained on one species to generalise. R2, on the other hand, requests clarification on a number of methodological details, clearer explanation of related work and also identifies a number of typographic and grammatical revisions.

Whilst the work presented has potential, the manuscript submitted appears to include a number of typographical errors and missing cross references (presumably reported by latex as compilation errors). Both reviewers also note repeated phrases appearing in different sections. These issues significantly affected the quality of the submission.

At this stage I recommend the following:
1. Carefully evaluate whether the methodological concerns identified by the reviewers are indeed valid and if necessary conduct further experiments with an appropriately revised protocol.

2. Improve the clarity of the methodological descriptions taking into consideration the reviewer's concerns regarding the reasoning presented concerning why models trained with 'balanced' vs 'unbalanced' datasets have different test set performance.

3. If possible, properly justify your arguments regarding the potential utility of TSS predictions from CNN models as compared with other TSS prediction approaches for cross-species predictions: ideally you should briefly review the prior work on this topic in the introduction. Bear in mind here that since the manuscript has been submitted to PeerJ CS, any discussion of the differences in biological mechanisms of TSS recognition across species will need to be explained in a way accessble to non-biologists. If, however, there are unique advantages of CNN that might be exploited for the cross-species prediction problem then it would of course be great if they were clearly presented (ideally with appropriate citations).

4. Follow PeerJ's guidelines regarding
i. citation of publicly available datasets: please provide precise URLs and dataset version numbers for the data used, rather than just the main download pages (eg. https://hgdownload.soe.ucsc.edu/downloads.html includes annotation and genomes from a large number of organisms).

ii. versioning of source code and dependencies - e.g. create a version tag for the cited github repository to ensure that the precise version of the accompanying notebooks are properly cited in the manuscript.

5. Consult colleagues or a proof-reading service in order to improve the readability of the manuscript. This will help future reviewers should you wish to submit a revised and expanded version of this study for publication.

Reviewer 1 ·

Basic reporting

The quality of the written english could be improved throughout the manuscript. Some areas of the paper are technical and need improvements in the quality of the writing to make them understandable. There are also duplications of some text in the methods section ("implementation" and "hyperparameter tuning" sections).

Experimental design

The authors do not describe how the TSS annotations used for training are collected. Are these TSS’s identified using cDNA clones, Illumina RNAseq, RACE sequencing/ CAGE sequencing of 5’ tags, GROseq/NETseq of nascent RNA, 5’GROseq of nascent 5’ tags, or PacBio/Oxford Nanopore long read sequencing, etc, etc. Many commonly used techniques, e.g. Illumina RNAseq, give very poor estimations of transcriptional start and termination sites due to tailing off of coverage towards the 5’ and 3’ of genes. The technique used to produce the annotations will therefore give some measure of the quality of the positively labelled training data. I expect that for a well annotated species such as human the TSS positions will generally be good, but this is not always the case.

The authors implement a data augmentation strategy, namely the sliding of the sequence window relative to the positive and negative examples. They report that the model trained with data augmentation performs better on the test dataset but worse on the held-out chromosome 21 test. This sounds like an example of data contamination/leakage – if the authors performed data augmentation before splitting the data into training and test data, there will be examples in the test set which are almost identical to those in the training set. The authors in fact allude to this themselves on line 294 - "The [data augmentation] approach is biased by the fact that similar sequences, albeit with different offsets... are found in both training and test sets". This makes any evaluation of the model on the validation or test set worse than useless as the model can achieve perfect scores by overfitting to the training data. Data augmentation should always be performed after data has already been split into training, test and validation subsets, to prevent contamination. I think that this need to be corrected as it is a major experimental flaw.

The authors apply models trained on human TSSs to other vertebrates (mouse, chicken, zebrafish). I would argue that training a model on one species before applying it to another is not a sensible proposal, since there may be unexpected differences in the sequence composition of TSSs in other species which lead to spurious results. To take an example from computer vision, you would not train a convolutional neural network to recognise dogs using only pictures of labradors, and then expect it to correctly identify pictures of poodles, since this is outside the bounds of the training data.

Validity of the findings

The authors suggest that using imbalanced training datasets is a good way to prevent model over-prediction when applied to the genome, where the positive class is rare. Will simply choosing a more stringent prediction threshold produce the same result?

The data used to train the model are not provided but are publicly available. Code used for training models is provided via github.

Reviewer 2 ·

Basic reporting

Regarding the Abstract, I would recommend to further elaborate the following two sentences for clarity purposes: "These approaches rely on the generation of sequences batches with known annotations for learning purpose" and "This cross-species application of convolutional neural networks trained with genomic sequence data provides a new technique to annotate any genome from previously existing annotations in related species". I would also discourage the use of "optimal" to describe the proposed strategy cause CNN offer an optimization that is not guaranteed to be a perfect/optimal formula.



Regarding the Introduction, while sufficient context is provided, the first half of the section would benefit from additional references to support the numerous messages provided.

Line 41: CNN are here introduced so the authors may want to rephrase the sentence saying something like "In particular" instead of "indeed".

Line 43: "illustrate [...] reliably" may be considered a stretch - CNN attempt to approximate an unknown function/mapping. Overall, the sentence would probably benefit if further elaborated to clarify the message the authors aim to deliver.

Line 61: "Firstly, we do not take into account prediction scores obtained from test sets initially retrieved from the training sets as a quality measure" is quite confusing. Do the authors mean "prediction scores obtained [...] *like(/in a similar fashion as)* the training set"? Test sets examples cannot be obtained *from* the training set.

Line 67: "We then propose a new application of the method" - which method? No method seems to be presented this far.

Line 74 (and 100): "human regions containing TSS" instead of "human TSS containing regions" may improve clarity.



Regarding the Methods, this section would benefit from additional references, e.g. no reference is provided for RefGene and all the data sources, sub-sampling and data augmentation, and the genome-wide standard score.

Line 82: the authors introduce the "positive *input sequences*" while they talk of "negative *class*" in line 87. This is somehow confusing.

Figure 1: the one-hot encoding does not match the encoding outlined in line 97. The dropout labels are represented before the max-pooling layers, is that correct? Why are dropout layers represented as labels instead of layers/masks? Isn't the size of the widest max-pooling layer = 64 (instead of 128)? Also, the figure shows only 1 input sequence although multiple annotations are represented in the output. More importantly, the convolutional layers appear to be arranged in parallel while Table 1 and 2 describe a more standard "pipeline-like" arrangement.

Caption of Figure 1: the last 2 sentences may benefit from further elaboration. The first one does not specify that the model has to be applied multiple times to give probabilities "over the full chromosome length" while the second one might contain too many details, e.g. the average z-transformed probability hasn't been introduced/mentioned yet.

Lines 102 and 106: it is confusing to talk of "layer kernels" or "convolution operation" instead of "convolutional layers" - which would also keep consistency with the paragraph.

Equation 1: j is not initialized.

Lines 117, 183 and 203, and caption of Figure 1: there is a refuse "over-fitting".

Lines 117-121: the authors may want to move these lines to the end of this subsection, after the concluding passage describing the model adopted, i.e. the fully-connected layer, etc.

Line120: dropout is typically applied after the non-linearity and pooling layer and not after "each layer" (in a CNN). Please clarify also whether dropout is applied after the fully-connected layer as well.

Line 136: the authors may want to specify that training on a GPU is typically faster than on a CPU.

Line 145: a full stop appears to be missing; the network architecture is detailed in Table *2*.

Line 146: no data appears to be on GitHub, only the program and respective utilities.

From line 150 onwards: there appears to be an issue with labeling the different (sub)sections as in "(see Section )".

Lines 148-162: this subsection may be better positioned after subsection "Conventional performance measures".

Lines 164-175: it is a copy of lines 135-146.

Lines 177-178: the authors may want to specify the unit of measurement, i.e. bp.

Lines 184-185: the authors appear to jump on "imbalanced data issue" from "training a CNN model on the balanced dataset" [line 177]. This creates several inconsistencies in the following subsections as well.

Table 1: "The first column" does not seem to "represents [only] different kernel sizes for 3 convolutional layers". Kernel sizes for 3, 4 and 5 convolutional layers are listed.

Line 222: a parenthesis is missing.

Line 227: the authors may want to support the "wide" with some references or, possibly, merge the line with the following one.

Line 239: AUROC and AUPRC have not been introduced in "Conventional performance measures" so it would be sensible to introduce here the acronyms.



~~Results
Line 246: it appears to be non-sensical since the predictive model is generating the predictions, i.e. the result of its attempt to differentiate positive and negative TSS.

Caption of Figure 2: please introduce the acronym "GC". It may also be beneficial to specify that the "30 trained CNN" are trained on different datasets and are not an ensemble.

Line 261: it may be beneficial to specify the value of Q for the 100* dataset. A similar explanation may go in line 91 as well.

Lines 272-273: this information may be more appropriate just in Caption of Figure 2.

Line 284: the authors may wish to clarify whether they talk about test set examples and training set instead of "test sets" and "learning examples", respectively.

Line 285: there is a refuse "Figure Supplementary Figure".

Line 300: the AUROC is typically defined as the "Area Under the ROC" and not as the "Area Under *an* ROC *Curve*".

Supplementary Figure 4a: the authors may want to specify that different *training* datasets are represented along the x-axis (same for Figure 1). The authors may wish to clarify whether they applied the same weighting scheme used at training time to calculate the entropy.

Line 302: "PR is" *not* "considered as a trustworthy measure [...]", is it?

Line 311: Binary Cross Entropy is the loss function chosen by the authors for this work, back-propagation works with other loss functions as well. Please clarify this.

Line 314: a perfect model is arguably a model with log loss = 0. What about generalization and overfitting? What *if* the dataset contains noisy information?

Lines 320-323: the authors may want to say that what they describe is intrinsically linked to the loss function adopted at training time, i.e. for this manuscript, the entropy discussed a few lines before.

Lines 324-328: the authors may wish to further elaborate to clarify the messages they aim to deliver. It may be sensible to also identify and label the test set used.

Line 330: this sentence is confusing as the authors are talking of "training and validation sets". Moreover, the only MCC scores reported on the test set (S_Figure4) disagree with this line.

Line 335: please clarify whether Zg corresponds to "the standard score". Same in Caption of Figure 3.

Lines 336-338 and 342-344: these lines appear to describe Figure 3 instead of "Comparing 1* and 100* models over a full chromosome". Thus, it may be sensible to move these lines to Caption of Figure 3.

Line 355: the authors may want to specify that the t-RNA cluster is highlighted in green *in Figure 3a,b*.

Line 375: the authors may want to clarify that "the signal to noise ratio is improved in the 100* models" with respect to the 1* models.

Line 384: the authors may want to refer to Figure 3 along to Figure 2.

Lines 395-396: it appears like an overstatement as some signal appears to be present. Please clarify.

Line 401: Down, T. 's PhD thesis appears to be unavailable. The authors may wish to cite an easier-to-access manuscript co-authored by Down T., e.g. DOI: 10.1126/science.1112014, or other works.



~~Discussion
Line 432: the cited manuscript [Wainberg et al.] claims that "there are now over 1 million genome *datasets*". Please clarify.

Line 434: the authors may wish to further elaborate this sentence and talk of "majority class" (or "most represented class") for clarity and consistency with the following lines.

Line 448: citation(s) required for "many genes are homologous".

Experimental design

~~Introduction
Line 45: the sentence appears to be vague, some additional detail or possibly reference may be missing. Do the authors aim to highlights the limits of current approaches, e.g. training and test sets do not cover the entire genome (which is not clearly stated at present)?
Moreover, the following lines do not seem to specifically address how the "training sets are obtained", e.g. line 47 mentions "Prediction" while line 49 talks of "Similar training datasets".

Lines 58 and 59: the authors may wish to further elaborate on the outcome and the differences of Kelley et al.'s and Jaganathan et al.'s works with respect to the present manuscript. How does this study fill in the current literature?

Line 91: the authors do not specify why the maximum considered value of Q is 100, e.g. why not 399?



~~Methods
Line 143: the authors may want to specify that they use a sigmoid function and, possibly(?), a softmax function in input and output of the dense layer, respectively. Do the authors use a sigmoid function (as in Table 2) instead of a classic softmax function in the output layer? Using a sigmoid function would result in a regression instead of a classification.

Line 158: it is unclear to me (1) how the authors calculate an element-wise average of a vector and (2) how such operation results in a new vector.

Lines 175-176: the authors may want to specify how many folds they used and what share of the data available. However, this information does not seem to match with the splitting outlined in line 178. It is unclear whether the final models are trained in cross-validation or on the entire training set. Moreover, the authors may want to clarify whether the architecture detailed in Table 2 is the only one used in section Results.

Line 192: when a = 0.5, equal importance is given to precision and recall; if a = 1, each TSS example will be weighted x times where x = (noTSS/TSS examples). Therefore, when a = 1, equal importance is given to "both classes". Please clarify.
Did the authors use this weighting scheme for any result reported in this manuscript?

Lines 206-206: do the authors ensure that the window sliding "around the TSS positions" contains TSS? Is the dataset still balanced after performing the described data augmentation strategy? The authors may want to specify that data augmentation has not been used to build all the datasets assessed in the manuscript and that the only method using DA is labeled as the "DA model".

Lines 216-217: FP, FN and TN are wrongly introduced.



~~Results
Line 270: the authors may want to clarify whether the positive TTS in the 30 1* datasets are the same.

Line 293: did the authors ensure that augmented examples stayed in the same split with the respective source examples when splitting the data in training, test and validation?

Lines 411-414: the authors may want to follow the same strategy previously followed to support the presumable superiority of more precisely annotated genomes. In particular, one chromosome or data not used at training time would give stronger results if used at test time.

Validity of the findings

~~Results
Line 240: the results of Umarov and Solovyev are on a different application. Please clarify.

Line 265: it sounds like an overstatement. Please elaborate.

Line 358: it may sensible to support this sentence citing some related work.

Line 406: the authors may want to give additional context about the choice of using a different r value, i.e. r = 400bp instead of r = 2kbp, for what essentially is one of the main messages of the manuscript.



~~Discussion
Line 450: the authors may want to add that the described methodology can *sometimes be applied* to more distance(/ different) species.

Line 452: it is unclear whether a more thoroughly annotated fish genome would do (as the authors state in the previous subsection).

Lines 454-456: these last few lines are essentially speculating on the possibility of applying CNN "to address the conservation of the [...] sequences" which is not validated in this manuscript.
From PeerJ's guidelines: The conclusions should be appropriately stated, should be connected to the original question investigated, and should be limited to those supported by the results. In particular, claims of a causative relationship should be supported by a well-controlled experimental intervention. Correlation is not causation.

Additional comments

The authors present a study on the potential of applying CNN on different species to predict genome annotations. The manuscript is quite comprehensive and some effort on improving the presentation would benefit the overall study performed.

---

## Round 0.2 · Minor Revisions

On behalf of both reviewers as well as myself I would like to thank you for your revised manuscript. Whilst a smattering of minor issues remain we are all in agreement that once these are resolved your work will be a well-received contribution to the field.

Both reviewers comment that some of the English in the manuscript could be improved. Please consider both my suggested typographic revisions (in the annotated PDF) and those of the reviewers.

Please also review the instructions to authors regarding use of species names, table formatting and figure legends, and also regarding citation of data sources (e.g. the range of genomic data and annotation in Methods - see comment in PDF).

We have requested a number of details be clarified in the text - mostly concerning specifics which are not immediately obvious to the reader, but also important methodological details which do need to be clearly communicated.

I have recommended revisions to figures and figure legends in order to improve their clarity and comply with PeerJ style. I also recommend some data be omitted (GC content in figure 2) - please look at comments in the PDF for detailed instructions. In general please ensure that any tools and scripts used in the creation of figures are properly cited and/or included in supporting information to maximise reproducibility of figures and results.

In addition to the scientific and methodological questions raised by reviewers, I highlight the following scientific issues:

1. Reviewer 1 notes that GC content across chromosomes is not constant and that local Z score may correlate with GC content (an observation that was perhaps intended by inclusion of the GC content plot in Figure 2). As a corollary to reviewer 1's question: could an alternative explanation for the different values of lambda observed for different levels of CpG count for Mouse/Human X chromosome predictions be that there is an imbalance in the distribution of CpG enriched transcriptional start sites across the training set ?

2. Different values of r have been used during the various experiments, but whilst the effect of adjusting r is described no discussion is given as to why different values were used. Why were different values used ?

Reviewer 1 ·

Basic reporting

The paper is much improved from its previous submission. In places the English could still be improved slightly. However, both the biological and computational concepts are explained with much more clarity.

There have been recent relevant publications which should now be cited, namely: https://doi.org/10.1038/s41598-020-57811-3

I think that perhaps the term "promoter" better (and more canonically) describes the sequences that are being predicted than "Gene start site".

Line 183: The model was tested on chromosome 21 which was withdrawn from the training set. This should be stated in the methods.

Figure 3: What are the green and yellow shaded areas on figure 3A and B heatmaps?

Experimental design

Line 210: The authors state that the performance of the model varies over different regions of chromosome 21. Human chromosomes are not homogeneous in the sequence content and have isochores with large differences in GC content. It would be interesting to see whether plot of local Z-score across a whole chromosome follows the isochores/GC-content of chromosome 21.

The authors have missed an opportunity to interpret the sequences identified by their model. Attention and saliency scores can be used to identify sequences that are important for predicting promoters. These could be compared with known sequences in e.g. the JASPAR transcription factor motif database. I do not think this needs to be done for publication, however it would improve the interest of the paper.

Validity of the findings

The authors have done a better job explaining the pitfalls of training on one organism and applying to others.

Reviewer 2 ·

Basic reporting

Although the manuscript has been improved considerably, the help of a native speaker may considerably improve the overall clarity and make the present manuscript even more professional.

Line 15: "learning [..] sets" is an uncommon way of referring to training sets.
Line 32: "in other species"? The message is unclear, please clarify.
Line 37: please extend on why "these annotations [...] cannot be determined experimentally for every organism and every cell".
Line 41: the authors may wish to specify that ML has been particularly successful "in Genomic applications" as well.
Lines 57-59: the authors may wish to cite [24].
Lines 65-66: did the authors attempt to reimplement [18] or analyze how much memory is "lot of memory"? It may be interesting to give a bit more context on the benefits and disadvantages of [18], i.e. to adopt windows covering the whole genome.
Lines 103-112: it is somehow confusing to interchangeably refer to human or mouse genome - sometimes without stating it but only the number of sequences.
Line 135: the authors may wish to cite the relevant manuscript descriving the back-propagation algorithm.
Line 149: the authors may wish to cite Keras as outlined at https://keras.io/getting-started/faq/#how-should-i-cite-keras.
Line 178: please clarify what "the input data" are. Does it refer to the training set or also the validation set? This is clear only after reading the Supplementary material.
Line 184: the authors may wish to clarify whether chromosome 21 has been withdrawn from the training *phase*, i.e. the validation set as well as the training set.
Lines 252: the authors may wish to support the former statement referring to the literature.
Line 257: the authors may wish to support the following statement "The two chromosomes carry homologous genes" citing [32].

Experimental design

Line 112: the authors may wish to introduce the sets used for the zebrafish and chicken genome.
Line 120: s is not explicitly introduced; does s have the same value for any kernel? The authors may wish to introduce the stride number adopted.
Lines 131-133: the authors may wish to specify that what they adopted (and describe) is a softmax layer.
Line 171: there is a discrepancy between "r kbp" and how r is introduced, i.e. r= 2000 bp or 400 bp (line 164). Also, the authors may wish to outline the equation to calculate λ.
Table 1: please clarify why the output shape of the first Conv2D is 32x284x1 instead of 296, i.e. b - s +1 = 299 - 4 + 1 = 296.
Line 181: the authors may wish to report the results they obtained on the mouse genome as well.
Lines 233-245: the authors may wish to specify the λ obtained in the four randomly selected GSS.
Line 284: the authors may wish to introduce a baseline training a model on the 4 sets of GSS curated, i.e. human, mouse, chicken and zebrafish, or building an ensemble using the 4 CNN, or selecting the best n sets/CNN, e.g. an ensemble using only the human and mouse genome.

Validity of the findings

Line 253: the cited work appears to be preliminary since the mammalian test is based on "313 sequences, of which 50 were kept aside for test purposes" [28]. Please clarify this in the manuscript, and possibly expand on the implications.

Supplementary material: 100* appears to be the best model in figure 4 see, although the authors say "the CNN model applied on the balanced data (1*) yields the best performance on the test set regarding Receiver Operating Characteristic curve (ROC)". Please clarify.

Additional comments

I would like to thank the authors for carefully addressing my comments and those of the other peer-reviewer. I believe that the manuscript has been considerably improved since the first submission and, thus, I congratulate the authors for the tremendous amount of work.

---

## Round 0.3 · Minor Revisions

I was pleased to see your revised manuscript - and must apologise for the delay in providing my (hopefully final!) review comments below. The majority of revisions listed are purely typographic/English language issues, but I have also explicitly noted clarifications and revisions that will improve the standard of reporting and precision of scientific language used in your paper.

1. l64-67 - the authors highlight the disadvantage of a methodology described in ref 18 as 'requires a lot of memory', but then go on in line 67 to say that the methodology proposed here is inspired from this last study. Reviewer 2 highlighted this in the previous version, and in your rebuttal you state that your method doesn't require expensive or highly specialised hardware (beyond a commodity GPU): however, leaving the statement in the manuscript *as is* doesn't help the reader understand this. The reviewer's point here was to ask you to explain in the manuscript why [18] was informative for this work: therefore you need a short additional sentence to explain why the approach they describe is relevant, despite its costly memory requirements.

2. l.85. Please include systematic names when you first mention the 'fish' and 'bird'. Formally you should also include the canonical citations for both these genomes at this point. You should also be consistent when you refer to these later - e.g. rather than bird simply say Chicken.

3. l. 89-102. I recommend here that rather than include the download URLs inline in the text you create a table listing each URL along with was downloaded and when (if you cannot instead identify the version number or publication date for each dataset), and change the text to refer to the table as follows
"Genomic sequences were downloaded for the reference genomes for Human (hg38), Mouse (mm10), Chicken (gg4) and Zebrafish (dr10) via the URLs in Table 1. Similarly, GSS positions for each genome were extracted from their respective NCBI RefSeq Reference Gene annotations (RefGene)."

4. Please address any discrepancies between the versions of figures & legends inserted into the main text of the manuscript and the ones appended at the end. e.g. the second version of figure 3 lacks a legend, and the CG content colour key in Figure 4 appears in different locations in the two versions of the figure.

5. Line 307-308: Use of the word 'activity' here is imprecise: the activity is the same (detects start sites) but their substrate specificity (ie the patterns they search for) are different: to avoid confusion, it is better to be explicit: "All together, these results suggest that the molecular machinery that interprets the genome sequence in order to find start sites of genes has a similar specificity in human, mouse and chicken, but a different specificity in fishes.".

6. Line 335-336: 'cannot be efficiently predicted' - you need to be more precise: 'cannot be reliably predicted with models trained on mammal and bird sequences'.

7. Similarly, to 5: Line 337-338 - I suggest here you revise to state this specifically: e.g. "further developments of our method may provide a new tool to quantify more complex patterns of similarity between different organism's nuclear machinery that interprets DNA sequences in vivo".

8. Grammatical corrections
G1. l.34 - delete ', notably': "of these non-coding sequences has been increasingly achieved through improvements in the"
G2. l.40 - 'An' should be 'A' - (there are many cases of 'An' which should be 'a' in the manuscript)
G3. l 42 - CNNs - *plural*
G4. l 47 - comma after enhancers : "predict enhancers, which are".
G5. l 50 - remove 'ing' from 'splicing': "or detecting splice sites [16,17]"
G6. l 73 - delete 'is' '..show that this tuning [is] has..'
G7. l 79 - delete 'an' 'rely on homologous molecular machinery'
G8. l 109 - remove space after GSS and before '.'
G9. l 139 - a not 'an' - 'may reach a local optimum'
G10. l 168, 169 - 'lines' in matrices / tables / 2D arrays are more commonly referred to as 'rows' (see also legend of figure 4).
G11. l 170 - insert 'all' - "..in a 10kb window around all GSS".
G12. l 190 - remove 'inductive rules' - "fails to generalise [inductive rules] to the new examples."
G13. l 193: alleviating to 'alleviate' : '.. to the balanced dataset to alleviate the..'
G14. l 202: change 'much less' to 'fewer' '.. false positives than..'.
G15. Figure 3 legend: fix typo 'oGSSn'
G16. l 207-9: the common English colocation is 'for the sake of completeness' (rather than exhaustiveness). Suggest revising the sentence to:
For the sake of completeness, the performance of more models (1*, 10*, 20*, 30*, 50*, 100*) evaluated using conventional metrics on test sets derived from the initial sample sets can be found in Supplementary materials.
G17. l 228 - add ',' after respectively : '1* and 100*, respectively, and \lambda ..'
G18. l. 252 - homologies->homology, and add an additional comma: "Mammalians show a substantial degree of homolgy in the DNA sequence found at GSS, and earlier"
G19. l 286 - revise 'similarity between the two mammalian genomes' to : "similarty between their genomes[32].".
G20. l 305. insert 'of' after upstream: 'are found upstream of many GSS'
G21. l 310-311 - This final opening sentence needs some polish: : suggest to avoid alliteration (millions used twice), consider replacing 'millions of gigabases' == 'petabase' .. e.g.:
"With the surge of DNA sequencing technologies, over a million genome datasets are now available and
petabases of transcripts are sequenced every year to annotate these datasets with functional marks [33]."
-- note - please check I have not changed the facts to fit better with the text here !
G22. l 315: remove 'the" - '.. genomic sequences is [the] unbalanced data, a'
G23. l 320: suggest 'impacts' rather than 'deteriorates' ".. the minority class (i.e. true GSS) impacts the learning process'
G24. l 324-326: explicitly state the name of the approach you've described "limited unbalanced datasets", and explicitly state that the best performing imbalance ratios you found are for GSS recognition:
"Using this approach, which we call "limited unbalanced datasets", we show that learning on imbalanced datasets can be performed effectively, and that for GSS recognition, a ratio of 1 to 30 positive over negative examples is usually sufficient to achieve a good signal to noise ratio in the prediction."
G25. l 329-330: "We use [the X chromosomes of] human and mouse GSS as [a] case study, and evaluate performance for models trained on each organism's other chromosomes."
G26. l 332. Delete comma : '..recover GSS in mouse/human with a model trained in the other organism[,] suggests'

9. Figure legends - during proofing please ensure figure legends do not include untranslated latex commands / symbols!

---

## Round 0.4 · accepted · Accept

Thank you for considering my revision requests in the previous round. I am also very flattered to be included in the acknowledgements!

A final read of the manuscript only revealed:
1. Line 342 - the 'a' is superfluous - "use as examples chicken and [a] zebrafish".
2. Supplementary information. I note you have not found a reference describing the efficacy of Matthews Correlation Coefficient for imbalanced classification problems: a detailed analysis on this topic was published by Chicco and Jurman (2020, https://bmcgenomics.biomedcentral.com/articles/10.1186/s12864-019-6413-7 ).